# What Inspiring Elements from Natural Services of Water Quality Regulation Could Be Applied to Water Management?

**Magali Gerino** [1,*] **, Didier Orange** [2] **, José Miguel Sánchez-Pérez** [1] **, Evelyne Buffan-Dubau** [1] **, Sophie Canovas** [3] **, Bertrand Monfort** [4] **, Claire Albasi** [5] **and Sabine Sauvage** [1]

1. Laboratoire Ecologie Fonctionnelle et Environnement (LEFE), 118 Route de Narbonne, CEDEX, 31062 Toulouse, France
2. UMR Eco&Sols, University Montpellier, Campus SupAgro, Bâtiment 12, 2 Place Viala, CEDEX 2, 34060 Montpellier, France
3. Modis, 7 Boulevard Henri Ziegler, 31700 Blagnac, France
4. Ceebios (SCIC) Centre D'études et D'expertise en Biomimétisme, 62 Rue du Faubourg Saint-Martin, 60300 Senlis, France
5. Laboratoire de Génie Chimique (LGC), CNRS, 118 Route de Narbonne, CEDEX, 31062 Toulouse, France
* Correspondence: magali.gerino@univ-tlse3.fr; Tel.: +33-625-690-955

**Abstract:** Theoretical and functional ecology is a source of useful knowledge for ecological engineering. The better understanding of the natural service of water quality regulation is now inspiring for optimization of water resource management, restoration and bioremediation practices. This transfer with a biomimicry approach applies particularly well in the urban, rural and agricultural areas, but is yet underexplored for water quality purposes. This natural service intensely involves the benthic boundary layer as a biogeochemical hot spot with living communities. A selection of processes related to the bioturbation phenomena is explored because of their influence on properties of the aquatic environment. The applications are valuable in a range of fields, from water treatment technology to management of ecosystems such as constructed and natural wetlands, streams, rivers, lagoons and coastal ecosystems. This paper gathers the more obvious cases of potential applications of bioturbation research findings on the biomimicry of natural services to water practices. These include pollution pumping by bioturbated sediment, water column oxygen saving during early diagenesis of deposits under conveyors transport and conservation of macroporous as well as fine sediment. Some applications for constructed devices are also emerging, including infiltration optimization and sewage reduction based on cross-biological community involvement.

**Keywords:** biomimicry; water; natural regulation service; management

## 1. Introduction

In recent decades, the study of the relationship between biodiversity and ecosystem functioning (BEF) has made considerable a progress toward developing a predictive understanding of how biodiversity influences biogeochemical processes in natural habitats [1,2]. In parallel, two decades of natural service studies have reinforced the relationship between biodiversity and natural services, more particularly for regulation services [3]. A second generation of BEF research is now starting to apply these findings for societal well-being [4]. It is argued here that for the management and development of sustainable aquatic and terrestrial ecosystems, it is as important to understand the linkages between key engineer species or functional groups, and their related functions and services, as it is to focus on species diversity requirements for ecosystem stability [5–8]. Based on this research, it is possible to make theoretical predictions concerning the effect of species or functional groups on process rates, matter flows, bioremediation and resiliency of ecosystems [9–14]. Acquiring this knowledge is a source of inspiration for ecological engineering, where natural populations and communities are the engines of resilience for one ecosystem site. The

requirements for sustainable practices of renaturation and conservation are particularly obvious in aquatic ecosystems, where biodiversity is declining more rapidly than in other habitats [15]. This paper focuses on water resource sustainability as a further objective. The natural service that establishes the link between this biodiversity and water quality delivery is the self-purification capacity of aquatic systems, also called the water quality regulation service by the Intergovernmental Science-Policy Platform on Biodiversity and Ecosystem Services (IPBES). In its global assessment of ecosystem service (ES) trends, the IPBES depicted that in general, regulation services are more dependent on biodiversity than other provisioning or cultural services [3,16]. Since aquatic biodiversity is eroding, these services are consequently severely declining in water systems, with a more drastic trend for the water quality regulation service. The loss of this service may be explained by a late recognition of this service delivery, as the self-purification of aquatic systems has for a long time been thought to be a myth rather than a reality. With the new challenge of water security for humans and nature, the interest is rising, and recent studies [17–20] are now able to measure this service in the natural environment. In situ investigations about the biophysical compartment supporting this ES highlighted the influence of functional "hot spots" in aquatic systems where biogeochemical processes transform the natural and anthropic matters [20–22]. These compartments are instream hyporheic zones, and include nutrient spiraling [23,24] and sediments of wetlands, lakes [25] or marine ecosystems. These spots are assimilated to ecotones and biogeochemical reactors with major control over the free water quality in the overlying water column, especially in terms of total nitrogen, phosphorus, carbon and many other pollutant fluxes, such as heavy metals and organic micropollutants [26,27]. In the field of functional ecology, the habitats and biodiversity that underpin the natural services of water quality regulation have been studied [28–31]. There is still a long way to go before all the communities that are potentially involved in the processes can be fully identified, but the literature agrees on the need to identify the benthic communities of invertebrates and the microbial consortium as the main influencers of water purification in situ [1,2,30,32] as well as in artificial contexts [33], such as constructed wetlands [34,35]. The major phenomenon involved in the resilience of this benthic compartment in freshwater and marine environments is called bioturbation [36]. With the implementation of a functional approach to explore the biodiversity at the bottom of aquatic systems, there is now a large agreement that bioturbation controls the biological, chemical and physical properties (Figure 1) of not only the sediment but also the overlying water [37–41]. The living community that generates bioturbation is composed of some species well-known for their evident effects, such as fiddler crabs in mangroves or arenicole polychetes, which are often called ecological engineers (EEs) [42]. EEs change their environment from one physical state to another, and partly comprise the allogenic engineers. Because of the delay in the water regulation service description, the potential of the application of useful findings for a biomimicry pathway is still underexplored.

What are the different processes that could inspire potential practice designs for restoration or bioremediation of aquatic sites or water resources? This question has gained particular interest as water resources become limited and the areas available to support this resource become restricted, such as in urban areas.

The purpose of this paper is to highlight recent knowledge coming from BEF studies in aquatic ecosystems that inspire new practices for these systems' management. This paper was written to describe a selection of the most obvious links that could be established between BEF that underpin the water quality regulation service and management practices of aquatic systems. Bioturbation is considered throughout this review as a major inspiring process because it is the linkage between BEF and the service delivery. Bioturbation is a very prolific and promising technology for remediation, cleaning, management and recovery of environmental contamination caused by microbial activity [43]. By illustrating some examples of biodiversity involvement in the bioturbation of the benthic boundary layer and this service, this paper also supports the functional importance of this ecosystem compartment as a key part of conserving the aquatic system balance.

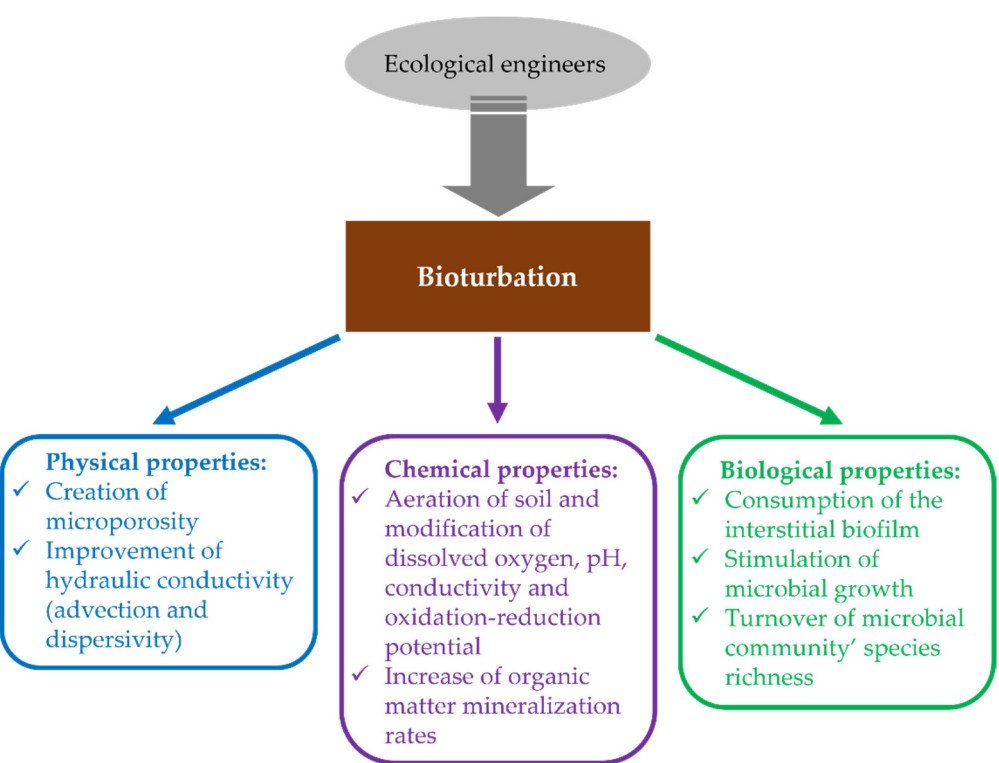

**Figure 1.** Biodiversity influence on porous medium properties.

Organism–sediment relationships are only part of the spectrum of processes which control the functioning at the water–sediment interface. Gardening, grazing, phytoextraction, biotransformation and bioamplification are among the many other processes identified as relevant for ecological engineering. These processes all belong to natural functions of the benthic compartment. Additionally, they can be amplified for environmental purposes in relation to water and soil quality restoration with natural resilience activation [44–46].

The method of transferring knowledge from nature observation to applications for societal problem solving is often related to nature-based solutions (NbSs) [47]. NbSs are just beginning to be developed in water management, and biodiversity is often poorly considered. Biomimicry refers to the identification of biodiversity and the processes involved in service delivery for improving water treatment and making them more sustainable. Aquatic biodiversity not only promotes the benefits of water quality regulation for people but is also a large source of good bioinspired practices that promote new low-cost technologies in this field.

Research could explore different strategies toward establishing sustainability of water management, but NbSs are certainly the most performant regarding co-benefits such as greenhouse gas (GHG) emissions, and climate change adaptation and resilience to extreme events [47,48].

## 2. Biodiversity Influence on Physical Properties of the Environment

### 2.1. The Mix of Sediments and Soils by Bioturbation and Pollution Pumping

There exist three major types of biotransports generated by the activity of the living fauna at the water–sediment interface. These biotransports are similar in the marine and freshwater environments [49,50]. The type of invertebrate effect depends on the sediment granulometry, with sediment biotransports being the major bioturbation processes in the wetlands and all fine deposits [51], as shown in Figure 2. In macroporous sediment of river beds, where particles are too coarse to be moved by bioturbation, the biofilm consumption by the invertebrate community and the gallery digging are the most efficient influences of invertebrates. In fine sediment deposits, the different water and particle biotransports

belong to three major categories: biodiffusion, biogeneration and conveying [49–52]. Biodiffusion is a mixing of the sediment in multidirectional ways so that the mixing effects lead to a homogenization of the surface layer when activity is intense. It works with adjacent exchange of particles, so it is considered a local biotransport compared to the other modes (bioadvection and biogeneration), which are non-local vertical transports [53]. Bioregeneration is usually associated with burrow digging and abandoning, producing fresh deposits at depth in the sediment, with a downward transport [37,54,55]. Oppositely, conveying is an upward transport of deep sediment toward the surface due to head-down-oriented worms that feed at depth and eject their fecal pellets at the sediment surface. The pellet accumulation at the sediment surface due to the large density of worms forms a new layer of refractory sediment and leads to a progressive subduction of the surface. In this way, the conveying process involves continuously renewing the sediment surface composition and pushing downward the new labile organic carbon that settles at the surface. This continuous burial of labile organic matter is generated by conveyor belt populations such as capitellids in marine waters [38,56–58] or tubificids in freshwaters [40,59–61]. This conveying process has been described as a major pioneering process that promotes colonization by other benthic organisms in the succession of the communities after a major perturbation event [36].

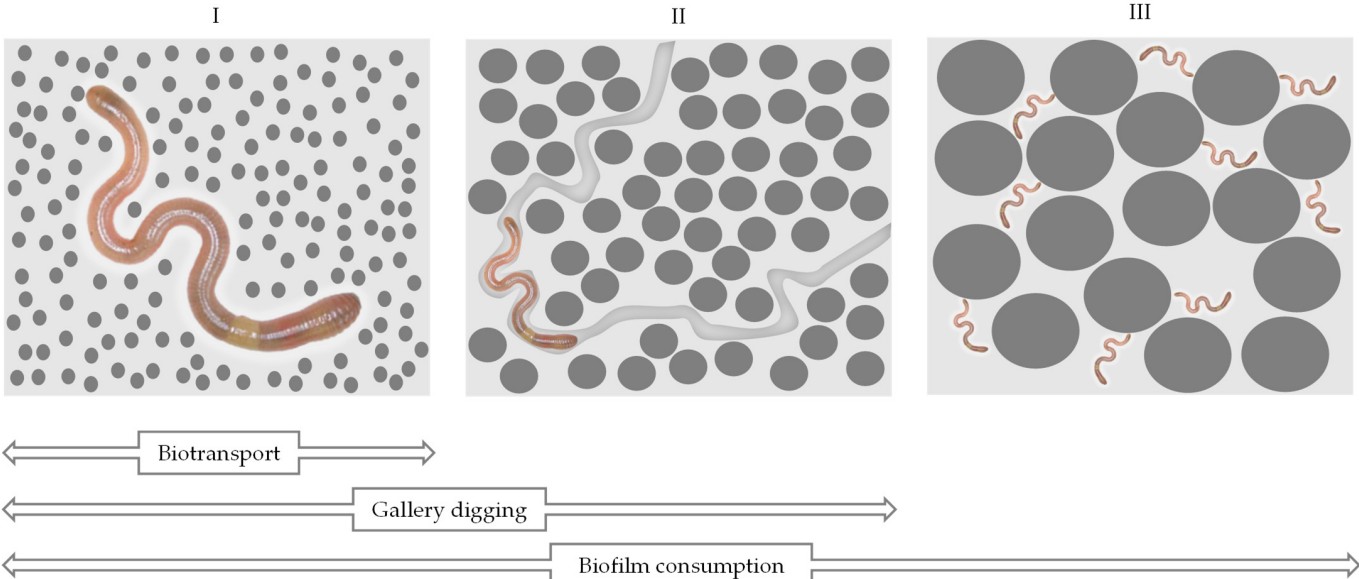

**Figure 2.** Three types of bioturbations controlled by the sizes balance between sediment/soil particles and engineering organisms. The arrows show that gallery digging and biofilm consumption are also possible with smaller granulometry.

This ability of bioturbation to capture new deposits and mix them into the sediment column applies to organic carbon as well as any other pollutants that settle down due to gravity, such as heavy metals. The new deposits are captured and subducted into the sediment column (called the benthic boundary layer) due to all types of biogenic reworking. Thus, sediment inhabited by abundant bioturbators is generally characterized by a higher level of pollution sequestered into the sediment [41,62]. This was shown by the records of radionuclides incorporation into natural sediment [63]. In recent deposits, $^{137}$Cs and $^{210}$Pb profiles according to depth show penetration that far exceeds the concentration expected from net rates of natural accumulation of sediment [64,65]. These profiles attest that more refractory particles, such as the labile particles, are also likely buried in the sediment.

If this relationship between intensity of bioturbation and stock of pollutants in the sediment explains the discrepancies that may exist between the quality of the overlying and the interstitial waters [66], it also reveals the benefits of having a living sediment for improving the water quality above the sediment in any aquatic habitats, as seen in ponds

and lakes. It has been estimated that conveyor tubificids increase cadmium fluxes from water to sediment by a factor close to two [40,67,68], as seen in Figure 3. More reasons why bioturbation should be promoted as an ecological engineering phenomenon of aquatic habitats are also given in part 4 of this paper.

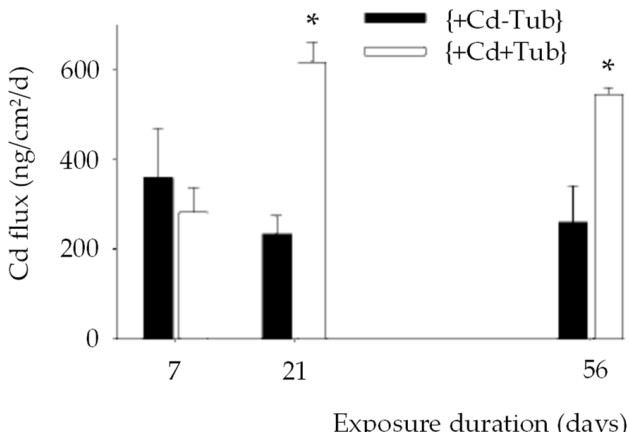

**Figure 3.** Estimated mean Cd fluxes from the water column to the sediment compartment of experimental units for the two experimental conditions: +Cd-Tub, water column contaminated with Cd and no Tubificidae in the sediment; +Cd+Tub, water column contaminated with Cd and Tubificidae in the sediment, over a period of time equal to 7, 21 and 56 d. Mean ± standard deviation (n = 3). * indicates a significant difference between the two experimental conditions ($p < 0.05$) [67].

This pollutant capture process is already active in freshwater lagoons and wetlands where it contributes to the well-known self-purification capacity of these ecosystems, with riverine wetlands often being compared to buffer zones for pollution trapping. This service of pollution pumping may also benefit constructed wetlands by artificially improving the bioturbation rates in the sediment.

Nowadays, the bioremediation capacities of natural and constructed wetlands are more largely described in terms of plant abilities. Phytoextraction, phytostimulation and phytodegradation are active in riparian vegetation of natural rivers and on riverbanks, as well as in planted filters. Phytoremediation is successful at removing both inorganic and organic pollutants from residues or soils [40,43,69,70]. However, very few studies have investigated the effects of bioturbation on the capacity of plants roots to capture pollutants from the sediment. A coupled bioturbation and phytoremediation experiment was previously run with a series of microcosms reproducing the water–sediment interface of wetlands and initial water contamination with cadmium [71]. The ecological engineers used were tubificid worms as a typical conveyor belt species and an aquatic plant, *Typha latifolia*, well-known for its extraction capacities. These tests showed that after 30 days under a bioadvection of 16 to 18 cm·year$^{-1}$ generated by the worms, the plant root bioaccumulation of cadmium significantly increased. Few other studies have demonstrated that aquatic bioturbation combined with phytoremediation is a more effective and alternative method of removing heavy metals by improving cadmium transfers from overlying water to sediment and then into the root system of plants [17,53,72]. In terms of ecological engineering, this observation suggests that the conservation or reinforcement of the invertebrates' communities in the soils and sediments, in addition to plants, should enhance the efficiency of the bioremediation strategies.

### 2.2. The Influence of Biogenic Structures on the Porosity of the Substrate

The interstitial biofilm is commonly defined as a community of living organisms attached to a solid surface and surrounded by a complex matrix of self-produced extracellular polymeric substances (EPSs). In all natural sediments and soils as well as constructed soils for water quality treatment, the accumulation of biofilm and organic deposits is one of the

drivers of porosity and infiltration capacities. The problem arises when biofilm growth at the surface of porous media (mud, sand and gravel as well as any artificial substrate), also known as biofouling, clogs the porous medium filter, reducing its filtration rate and compromising water purification, as seen in Figure 4 [73–78].

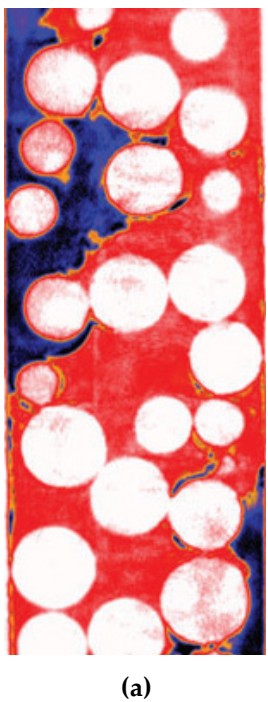 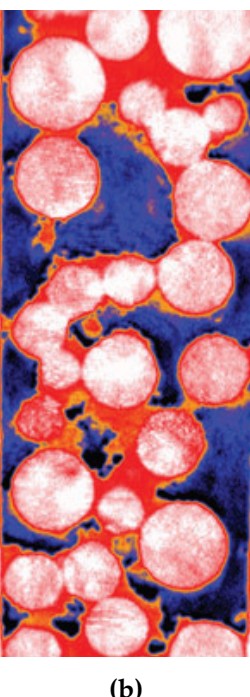

(a)                                                    (b)

**Figure 4.** Examples of reconstructed (X-ray Skyscan 1174 data) sectional slices for the entire length of the column obtained after 10 days at a flow rate of approximately: (**a**) Q = 0.07 mL·s$^{-1}$; (**b**) Q = 0.5 mL·s$^{-1}$ (a pseudocoloration has been applied to the images using ImageJ on the basis of the ceretec LUT and only for visualization purposes). The whitened parts correspond to the beads, the red parts to the biofilm and the dark blue parts to the aqueous phase [75].

Microbial growth increases notable changes in the effective porosity, hydraulic conductivity and dispersivity [79–82]. Soil bioturbation is associated with the production of soil macropores as burrows or gallery networks that influence numerous physical properties of the substrate such as water infiltration [83]. Biostructures (burrow networks) dug in the soil column as well as in sediment create an additional porosity that changes the saturated hydraulic conductivity $K_{sat}$ [84]. Relationships between total macropore properties and $K_{sat}$ showed that the most important properties explaining $K_{sat}$ were (i) the volume of percolating macropores, (ii) their diameter, (iii) the critical macropore diameter and (iv) the number of macropores [85]. This influence is quite relevant regarding the soil and sediment of the constructed filters as seminatural systems used for water treatments. Many types of filters are used for extensive devices as nature-inspired equipment that could be utilized on a sanitation site (individual scale) or in constructed wetlands, such as saturated or hybrid filters, aerated filters, etc. In water filters, the bulk hydraulic conductivity of the columns decreases by up to three orders of magnitude when the biofilm biomass reaches its maximum [86,87]. The addition of endogeic macrofauna in any of the treatment technologies based on water filtration could be an advantage for the filtration capacity of the filters (Figure 5). The continuous digging of burrow networks by earthworms is the most studied effect since these invertebrates are now well-recognized as ecological engineers [42,88–90]. The diversity of the species in the three functional groups of anecic, endogeic and epigeic reveals a range of solutions for improving the porosity in the entire soil column of the filters. The exploration of the optimal combination of worm species is still a research topic that would require further investigations.

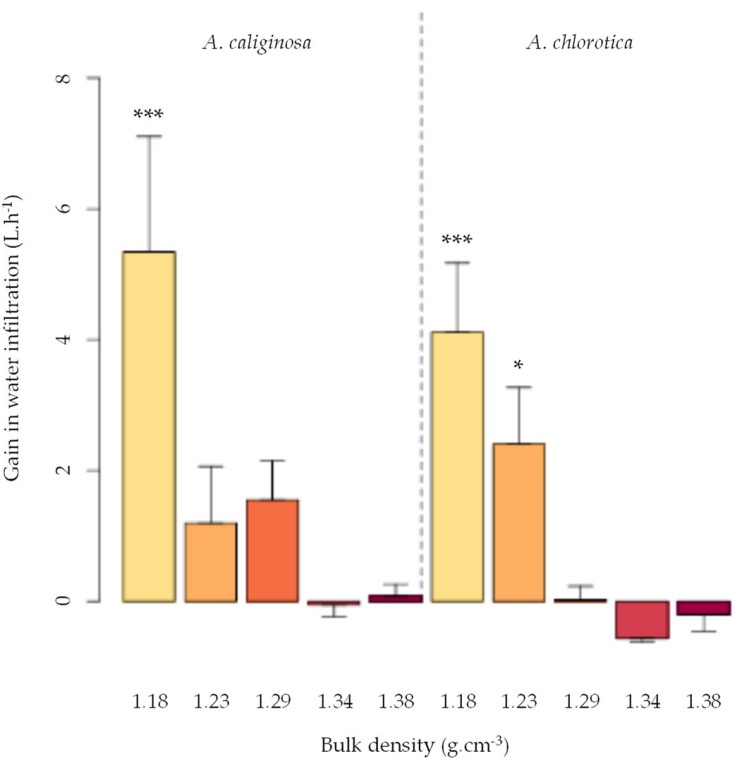

**Figure 5.** Gain in water infiltration (means + standard error) compared to control without earthworms when three individuals of either *Aporrectodea caliginosa* or *Allolobophora chlorotica* endogeic lumbriculids were incubated for 6 weeks in soil cores with increasing bulk densities. Stars indicate values significantly different from zero (each species was tested separately) [89].

## 2.3. The Influence of Biofilm Consumption on Sediment Permeability

In macroporous sediments, the consumption of interstitial biofilm by detritivorous invertebrates also changes the macroporous porosity and interstitial water flow through much deeper sediment columns [31]. In the marine environment, the bioturbated layer thickness ranges from 10 to 30 cm of the top sediment, whereas in freshwater, the invertebrates still occur in the hyporheic sediment at several meters under the river free water [39]. In these coarse sediments that characterize the river hyporheic zone, the biotransports are no longer efficient because of the size discrepancy between the particles and most of the interstitial invertebrates, but the bioturbation is active and favors the water fluxes into the interstitial media (Figure 2). In these conditions, bioturbation also acts through the consumption of the biofilm that limits the sediment clogging during long periods of low discharges. The addition of a population of biofilm consumers in the artificial media of filters appears as a bioinspired solution by analogy with the functional role of the interstitial community that lives in the hyporheic sediment of natural rivers.

If this solution could be applied in saturated filters, most of the technologies based on water filtration could work in non-saturated conditions. This type of environment drives the search for detritivorous engineers that live in semiaquatic environments for their further introduction into sewage water filters. An increasing number of studies are testing the effect of earthworms of the genus *Eisenia* on the sediment permeability, with the aim of introducing them into future filtration systems [91], but few of these studies have demonstrated their application in demonstration pilots or living labs. In parallel, there is a rising number of farms that are developing sewage filtration with the help of lumbriculids in systems called lumbrifilters or lumbricomposters, especially for those farms that produce vegetables by aquapony [92].

## 3. Biodiversity Influence on Chemical Properties of the Environment

Major bioturbator organisms are usually compared to ecosystem engineers as agents of biogeochemical heterogeneity creation. These organisms can affect biogeochemical reactions by changing the availability of resources for microbes (e.g., carbon and nutrients) or by changing abiotic conditions affecting microbial reaction rates (e.g., oxidation reduction potential and temperature) [4,32,93,94]. Through invertebrate and microbial relationships such as biofilm grazing and gardening, the invertebrates obtain the ability to influence the growth rate as well as the intensity and quality of biodegradation reactions conducted by the microbial consortium in the sediment [95] and the soil [96]. An increasing number of in laboratory and field experiments have demonstrated that invertebrate biodiversity influences retention of excess nutrients as well as different types of pollutants, such as trace metals [66] (e.g., Cd by bioaccumulation in sediment [67,68] and plants [40,97]) or organic pollutants (e.g., atrazine [71,98–100]). A first case study, the ANR (French National Research Agency) project called Imbioprocess, illustrated the influence of biodiversity in microcosms with and without invertebrates, and highlighted the effects of interactions between microbes, macro- and meiofauna on N-$NO_3^-$ in macroporous stream sediment [33,101]. After 56 days, N-$NO_3^-$ reduction rates in sediment (fresh weight) without invertebrates addition and after treatment with meiofauna and macrofauna grazing ranged from $3.76 \pm 0.35$ to $8.92 \pm 0.69$ mg $N \cdot d^{-1} \cdot kg^{-1}$, respectively.

A second type of example illustrates the potential of the hyporheic zone to eliminate $NO_3^-$ excess [102] and sewage-born micropollutants such as pharmaceutical residues [103,104] that flow in streams and rivers. Studies performed under experimental conditions copying streams and bank filtration with artificial groundwater recharges show that the same compounds considered as persistent in wastewater treatment plants (WWTPs) are transformed in river sediments or other porous matrices [105–108]. Indeed, the microbial consortium in natural environments is much more diverse than in WWTPs [103]. Thus, the hyporheic zone is identified as the key compartment for the degradation of persistent micro-organic pollutants (POPs) in rivers, which is also reinforced by the longer residence time of pollutants in contact with biofilm in this media than in free water. Then, the invertebrate community can have a favorable effect on biodegradation of the micro-organic pollutants by the microbial biofilm as it acts on the transformation of $NO_3^-$ excess. An interesting paper [109] explores the impact of the multiple-degrader community of porous media on the biodegradation of the river-dwelling organic compound phenanthrene. The application of this observation into water management practice shows the importance of conserving river bed sediment during the river management. In the river spiraling phenomenon, the water is forced to pass through the sediment with the help of the horizontal and vertical heterogeneities of the river morphology. In this way, it appears that the naturalness of the river with both its mineral and biological composition matter for the delivery of the service of water quality regulation, as well as the conservation of its original river bed morphology, play important roles in the functioning and pollutant retention. This should be particularly kept in mind for management of rivers and streams under human pressure, such as downstream effluent inputs.

## 4. Bioturbation Influence on Organic Matter Degradation and Oxygen Saving

The influence of bioturbation on sediment chemistry is often associated with its effects on interstitial oxygen gradients. This is a relevant variable of the sediment that controls the spatial trends of the whole diagenesis of the organic matter. Aller, as early as 1980 [110] and 1982 [111], and later several papers from other authors within a review [41], described how the different bioturbation types could change the oscillations of oxido-reduction potential, the organic matter degradation and the pollutant biodegradation.

Bioturbation promotes priming and total remineralization of sedimentary organic matter in multiple ways. An additional effect of the vertical transport of material by bioturbation is the injection of reactive organic carbon coming from the water column into deposits of subsurface horizons. During benthic fauna activities (feeding and burrowing), the labile substrate is brought into close contact with more refractory material at depth

in the sediment [41]. These biotransports are of particular interest when the objective of bioremediation of water is to reduce oxygen uptake from rich organic deposits of eutrophic systems or to reduce pollution at the water-sediment interface. The main effect of these conveyors is changing the balance between aerobic and anaerobic microbial metabolisms. The perpetual burial of labile organic matter coming from the water column into the refractory environment at depth favors anaerobic and anoxic degradation from partial to total diagenesis. Thus, the bioturbation process acts as an anaerobic stimulation of the organic matter degradation that will minimize the consumption of oxygen from the overlying water of the sediment. As an example of a potential application, *Capitella* sp. was co-inoculated with bacteria as an active conveyor in organically rich sediment to enhance the degradation of excess organic matter in sediment below fish farms [112,113].

The critical role of bioturbation to capture the labile deposit and to mix it into the inhabited sediment column is largely demonstrated in freshwater researches [52,67,114] as well as in marine studies. A few examples were provided by Long Island Sound and Great Peconic Bay research focusing on the rapid penetration, utilization and dissipation of fresh and reactive deposit from spring bloom inputs [38,41,115]. The understanding of this process for lagunes, wetlands, lakes and ponds management is fundamental for engineering practices orientation. Removing the fine sediment that covers the bottom of an aquatic ecosystem is often recommended for recovering the correct level of oxygen concentration in the water column. This is clearly a practice that removes the potential biological benefits of conveyors. By removing the sediment compartment where the benthic invertebrates and microbial community live in lakes, pounds, swans, etc., this practice eliminates the benthic part of most of the nutrients and carbon cycles, for example, where nitrate could be transformed into $N_2$ gas to be released into the atmosphere. Oppositely, the conservation of fine sediment and bioturbation potential would stimulate the natural resilience of the lentic ecosystem. A fine sediment layer should be maintained at the bottom of the water column for restoration of the bioturbation process by the pioneer invertebrates that are conveyors. The biotransport made by these engineers would favor oxygen saving in the deepest water. If bioturbation also increases the oxygen penetration in the sediment, this occurs in the bioirrigated galleries, with a local effect on the surrounding sediment (Figure 6).

Clearly, bioturbation during the first step of the ecological succession of living organisms in sediments and soils, produced by pioneer conveyors such as tubificid oligochaetes and chironomid insect larvae in freshwater or capitellids in marine water, is more likely to reduce aerobic metabolism than later successional steps. The sustainable practices of ecological engineering that should be implemented due to these observations should be sure not to cancel the primary process of the successional evolution of the benthic boundary layer. In the absence of bioturbation, the labile organic matter transformation is restricted to the first millimeter of the sediment with a metabolism rate that starts in aerobic conditions until the lack of oxygen obliges it to turn into anaerobic and then anoxic metabolism. When the benthic community succession proceeds, the thickness of the oxic sediment layer increases, and the balance between oxic and anoxic recovers. More generally, it is assumed that the total mineralization of organic matter in bioturbated deposits increases due to the increased inventory of reactive organic carbon as a function of biomixing rates [41].

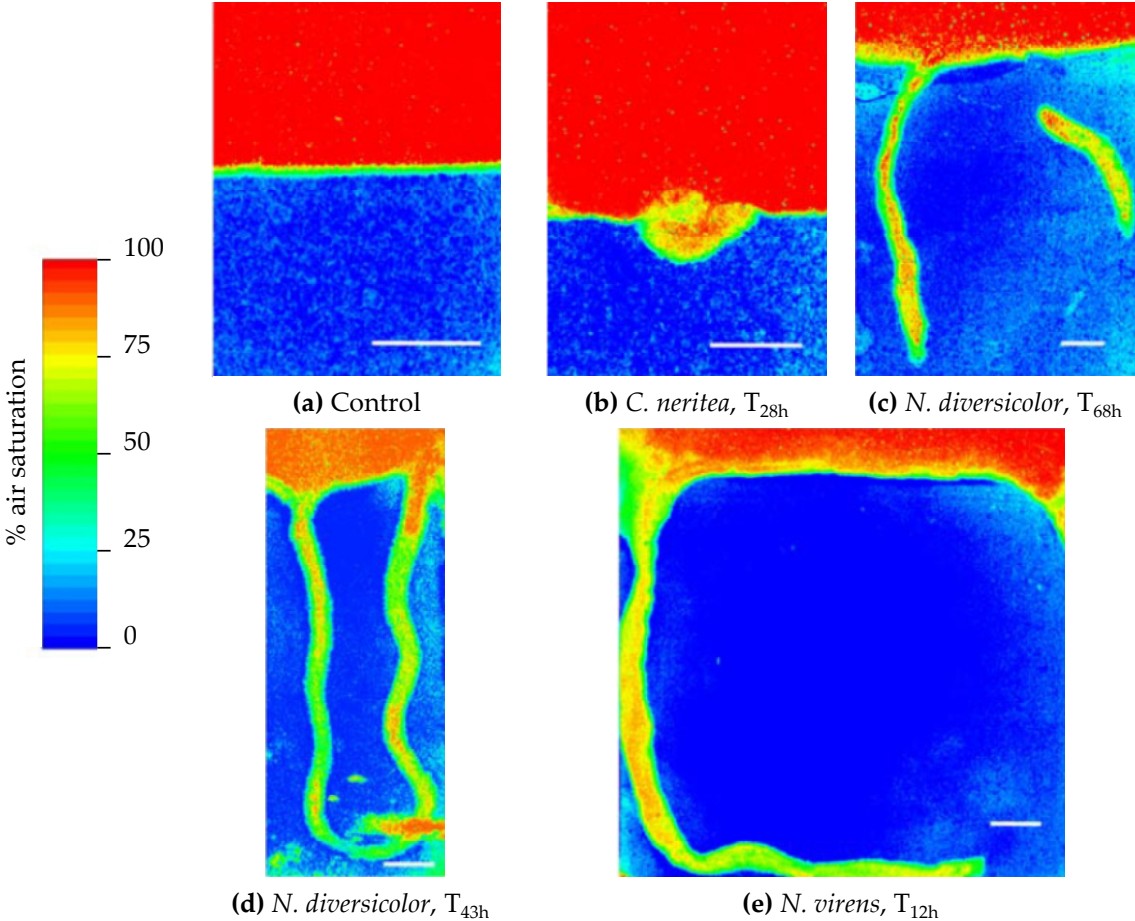

**Figure 6.** Examples of oxygen distribution in sediments. The gray-scale images were converted into false color. (**a**) Control without organism (resolution: 570 × 570 μm); (**b**) *Cyclope neritea* (resolution: 610 × 610 μm); (**c**) *Nereis diversicolor* (resolution: 500 × 500 μm); (**d**) *N. diversicolor* (resolution: 790 × 790 μm); (**e**) *Nereis virens* (resolution: 770 × 770 μm). Times at which the pictures were acquired are indicated [116].

## 5. Biodiversity Influence on Biological Properties of the Environment

Most of the processes that involve biodiversity for water treatment are so far focused on the manipulation of species richness within a single trophic level, usually plants or the microbial consortium. However, according to insurance and stability hypotheses of theorical ecology, the dynamic of a restricted set of species, and the ecosystem processes they drive in natural systems, cannot be realistically understood without reference to the dynamic of other species and processes within the same system at higher or lower trophic levels [117]. In other words, in the phytoremediation practices carried out to date, with plant involvement for soil restoration, the outcome of complex food web interactions between trophic levels has not yet been achieved. These cross-trophic level interactions (for example, between invertebrates and plants, or invertebrates and biofilm) have not only the potential to modify the metabolism of the ecological engineers, but also to change their population size, turnover, adaptation to stress, risk vulnerability and resilience. This theory supports several changes in agronomy practices, such as the development of agroforestry that integrates different species and crop auxiliaries to limit pest invasion. The application of this theory to water management is still limited, and demonstrations of its effectivity are missing. The co-benefits of recovery, from favoring the cross-communities' interactions to overlying air composition, are significant. Obviously, the type of greenhouse gas emitted differs from that produced by artificial sewage treatment and natural wetlands, which both degrade large amounts of labile organic matter. Nitrogen production is more complete in

natural environments where the microbial consortium is more likely to drive the whole set of diagenetic pathways [118–120].

## 6. Biodiversity Influence on Sewage Quantity Reduction

Other applications of the cross-communities' interactions are rising in the more artificial sector of water treatment. Since 2000, with a biomimicry approach from aquatic trophic web composition, authors have tested the coupling between water treatment and invertebrate populations for sewage reduction [121,122]. Some authors have tested this coupling in the framework of conventional pollution of domestic water (C,N,P) with sludge produced by activated sludge or membrane bioreactors [123–131], or with heavy metals in the water [132]. They demonstrated that the addition of a higher trophic level in the system of the treatment process is valuable, achieving a sludge volume reduction of up to 70% that comes together with better biomass aeration [133], and even a decrease in membrane fouling in the case of membrane bioreactors. The uptake by predation of the total solid particle mass ranges between 36 and 75% depending on the type of reactors or systems used for the demonstration.

This assumption has generated research that tends to demonstrate how the addition of a predator population in sewage treatment could help at reducing the quantity of slurry. These early findings support evidence for further investigation into water management, but certainly, there should soon emerge solutions inspired by this natural trophic balance knowledge to deal with slurry excess. The addition of higher trophic levels by biomimicry of the aquatic food web should help in balancing slurry excess in more natural devices compared to regular treatment with only active microbial biomass.

## 7. Conclusions

There is still a long way to go before adaptation of these natural processes to human practices is achieved, but a range of benefits are expected as an award for this investment. The demonstration of these benefits should help in the evolution from civil to ecological engineering. The potential of benefits of the evolution of practices inspired by benthic ecology is huge and still underestimated. The evolution from fully human-made technology to more nature-based solutions should bring a large range of co-benefits related to energy saving and low greenhouse gas emissions together with better stability and resilience. Biomimicry and NbSs are often referred to as innovative, but they should not exclusively include "new" solutions. Whilst the biomimicry concept offers new opportunities and brings added values, it also encompasses existing ideas, the current knowledge and interdisciplinarity approaches. With these demonstrations of potential benefits, this paper wishes to encourage the interdisciplinary research, innovations and actions for new biotechnologies and practice evolution. Reaching this understanding requires a supplementary effort to include new knowledge into current practices and technologies that could evolve within this approach [47]. The first generation of functional ecology research focused on one population or one trophic level and one process for maximizing its capacity with the first assumption that a simpler system would be more easily controllable. With the findings on BEF and the demonstration of the role of biodiversity in ecosystem stability, it became evident that multifunctional group assemblages are more performant for water-derived energy and matter flux management [134]. The knowledge is now available for a second generation of researches and demonstrations in water treatment and management that should emerge from involvement of additional biodiversity in more complex devices that match with the current demands for energy transition and more stable water production and recycling. The capacity to maximize the delivery of water quality services and simultaneously fulfill the specific needs of stakeholders will depend on the conservation and involment of the biodiversity, the identification of ecosystem service (ES) cascade (who is making what) and the acceptability of practice evolution.

The assumption that biomimicry and NbSs are not performant enough compared to fully man-made technology and artificial solutions is a myth. The benefits of biomimicry

require demonstrations at the pilot scale, which are more often referred to as NbSs, and that will favor its adoption by a larger range of stakeholders.

The challenges are more intellectual and related to change resistance than scientific obstacles. They require those changes coming from different academic traditions (ecology, civil engineering, chemistry, economy, water management, etc.) to fuse their key principles into a useful set that is comprehensible and accessible to all for cross-fertilization [47].

**Author Contributions:** Conceptualization, M.G.; investigation, S.S., D.O., E.B.-D., J.M.S.-P. and C.A.; resources, S.S., D.O., E.B.-D., J.M.S.-P. and C.A.; data curation, B.M.; writing—original draft preparation, M.G.; writing—review and editing, S.C. All authors have read and agreed to the published version of the manuscript.

**Funding:** This article partly benefits from the OBIOM project (Optimisation des filtres plantés par BIOdiversité augmentée: effet sur la Matière organique) publicly funded through ANR (The French National Research Agency) under the "investissements d'avenir" programme with the reference ANR-10-LABX-04-01 Labex CEMEB and coordinated by the University of Montpellier.

**Institutional Review Board Statement:** Not applicable.

**Informed Consent Statement:** Not applicable.

**Data Availability Statement:** Not applicable.

**Acknowledgments:** This article was written following an invitation to participate in the 2022 international GRUTTE Congress (https://gruttee2022.sciencesconf.org/ accessed on 9 September 2022). The ideas for this article come from outcomes and discussion within the SmartCleanGarden concept project (Research Award Convergences, from World Forum "Zéro Exclusion, Zéro Carbone, Zéro Pauvreté", Paris, 2018) and the new BioROC project (call WOc, "Eau'Occitanie challenge", 2022–2025).

**Conflicts of Interest:** The authors declare no conflict of interest. The funders had no role in the design of the study; in the collection, analyses, or interpretation of data; in the writing of the manuscript; or in the decision to publish the results.

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
