# Peer review of "What Inspiring Elements from Natural Services of Water Quality Regulation Could Be Applied to Water Management?"

_water, doi:10.3390/w14193030_

Round 1

Reviewer 1 Report

I rarely review articles, but I liked the title of the article and the first half of the abstract. It was a good and catchy title and beginning, but on further reading the article became questionable. The key problem with this article is the lack of a clearly stated goal. The existing wording "This review is gathering a selection of the knowledge that could be linked to interdisciplinary research, innovations and actions for new biotechnologies and practices" is too vague. This problem is also evident in the structure of the article. The introduction is devoted to describing the problems of BEF, NBS, and ES, but then the article is actually reduced to writing about the effects of bioturbation on various environments. The authors could prepare a good article on the application of NBS to water management, in which bioturbation would be one of the methods used. I don't think anyone would argue that the diversity of natural properties is much broader than just bioturbation. An alternative option would be to prepare an article specifically on the application of bioturbation to water management. This, too, is a good topic for research. The existing version of the article, which mixes both options, does not unfold the logical sequence all the way through and will frustrate the reader as he or she reads. 

Specific comments: 

1. What is meant by Nature regulation service, which is mentioned in the title of the article? It should be defined in the Introduction. 

2. The second half of the abstract is written very vague and unspecific, you need to make it more clear. 

3. Also need a clear definition of the term Ecological engineer, which is the first link in Figure 1. 

I gave the article an average rating, as its potential is much higher than the version presented. After defining the purpose of the article and building a new structure, it could be much better. 

Author Response

Response to Reviewer 1 Comments

Point 1: I rarely review articles, but I liked the title of the article and the first half of the abstract. It was a good and catchy title and beginning, but on further reading the article became questionable. The key problem with this article is the lack of a clearly stated goal. The existing wording "This review is gathering a selection of the knowledge that could be linked to interdisciplinary research, innovations and actions for new biotechnologies and practices" is too vague.

Response 1: We first would like to thank the reviewer 1 for this remark that invited us to precise more restrictively the objective of this paper, and to make it clearer since the beginning for avoiding any over expectation. The abstract was almost fully rewritten. In this way, the sentences about a general approach that could apply to any other transfer of knowledge from nature to biotechnology were removed. They were replaced by the general goal explanation (that is the use of the research fingdings about water quality regulation service sources for management practices evolution in aquatic environment for a review of some typical examples), in the abstract (lines 25 to 26) and in the introduction (lines 91 to 95).

Point 2: This problem is also evident in the structure of the article. The introduction is devoted to describing the problems of BEF, NBS, and ES, but then the article is actually reduced to writing about the effects of bioturbation on various environments. The authors could prepare a good article on the application of NBS to water management, in which bioturbation would be one of the methods used. I don't think anyone would argue that the diversity of natural properties is much broader than just bioturbation. An alternative option would be to prepare an article specifically on the application of bioturbation to water management. This, too, is a good topic for research. The existing version of the article, which mixes both options, does not unfold the logical sequence all the way through and will frustrate the reader as he or she reads. 

Response 2: Thanks again for this suggestion to shorter the BEF, NbS and ES statement and to focus earlier about bioturbation and benthic communities’ involvement in the water quality improvement. We confirm that the main goal (following the second option of the reviewer) is the application of bioturbation in the water management and several modifications were supplied to make this option more direct and evident. Among these changes:

  • The sentence “One of the last reports of the Intergovernmental Science-Policy Platform on Biodiversity and Ecosystem Services (IPBES) investigates how biodiversity influences biogeochemical cycles that regulate system productivity, respiration, and carbon storage” was removed;
  • The mention of aquatic systems is introduced earlier and added as soon as line 40 in the sentence “It is argued here that for the management and development of sustainable aquatic and terrestrial ecosystems, it is as important to understand the linkages between key engineer species or functional groups…”.
  • Some other general statements were moved in the conclusion as the sentence “The capacity to maximize the delivery of water quality service and simultaneously fulfill the specific needs of stakeholders will depend on both the conservation of this biodiversity, the identification ecosystem services (ES) cascade (who is making what) and the acceptability of practices evolution.”

In the introduction, we tried to give first a short history of the BEF and its progress in the recognition of the biodiversity involvement in ES. Then, the necessity of sustainable practices for aquatic ecosystems and its biodiversity is given as an argument to focus on these systems. A short state of the art about natural service of regulation of water quality (> line 50) and bioturbation are then introduced as a source of inspiration (> line 76). The references to the biomimicry approach applied to aquatic systems are given in lines 113-115 “NbS are just beginning to be developed in water management, and the biodiversity is often poorly considered.”, but most of the benefits of this approach by biomimicry and NbS were moved into the conclusion.

Point 3: 1. What is meant by Nature regulation service, which is mentioned in the title of the article? It should be defined in the Introduction. 

Response 3: Yes, the mention to all Nature regulation service was too large. The title was revised to focus on the water quality regulation service. All examples of knowledge transfer to practice improvement are referring to water quality.

Point 4: The second half of the abstract is written very vague and unspecific, you need to make it more clear. 

Response 4: The second half of the abstract was fully rewritten, in order to point out the main topics that are illustrated in this paper. A part of the answer to this point is detailed in the previous Response 1.

Point 5: Also need a clear definition of the term Ecological engineer, which is the first link in Figure 1. 

Response 5: The definition of ecological engineers (EE) as applied in this paper is given by Jones et al. (1994). In this definition, EE are organisms that directly or indirectly are modifying the availability of resources for other species, and other than themselves. In most of the examples explored in this review, the EE are allogenic that change their environment by transforming living and non-living material from one physical state to another via mechanical or other means. The best illustration of allogenic EE are bioturbators that changes the sediment from unmixed and homogeneous layer to sediment with galleries or mixed sediment. The precision “allogenic” and Jones et al. reference were added in lines 83-85.

  • Jones, C.G.; Lawton, J.H.; Shachak, M. Organisms as ecosystem engineers. Oikos 1994, 69, 373-386. https://doi.org/10.2307/3545850.

Reviewer 2 Report

I like the topic as it is considering the use of natural process managing water resources in terms of water quality and environment. Some studies on streams or bank filtration with artificial groundwater recharges show that the same compounds considered as persistent in WWTP are transformed in river sediments or other porous matrices. That is a very interesting finding and worth pursuing more public and user acceptance. The study would add values to the literature and the engineering and scientific communities. 

The experimental set up, types and number of experiments conducted for this study are not clear. It looks like a review article, but it is not clear. 

Page 7 of 16, Line 259: A citation in this line missing that is showing error.

Author Response

Response to Reviewer 2 Comments

Point 1: The experimental set up, types and number of experiments conducted for this study are not clear. It looks like a review article, but it is not clear. 

Response 1: Yes, this article is designed as a review of case studies of knowledge transfer from ecological sciences into local practices. To make this point clearer for readers and editors, we changed the Type of the Paper in line 1 from Article to Review. The starting hypothesis that underpins this review is that nowadays there exist a fair amount of knowledge that could be useful to consider more efficiently for ecological engineering progress. There is not a new experiment run by the authors that sustain this review. It is just a selection of examples that could be good candidates for this knowledge transfer. This is the objective of this paper that was added more clearly at the end of the introduction. The first part of this paper was also reworked to earlier focus on the investigations related to the aquatic systems restoration and management with the help of the benthic ecology. In this way, inspiration from the bioturbation process is introduced as a resourceful approach for damaged aquatic environments.

Point 2: Page 7 of 16, Line 259: A citation in this line missing that is showing error.

Response 2: We corrected this error: link to Gilbert et al. (2003) reference in the list. In addition, we added several other references in this paragraph like Biles et al. (2002), Kukwa et al. (2022), Le et al. (2016), Teal et al. (2013), Ciutat et al. (2005)Anschutz et al. (2012), Leveque et al. (2014), Farenhorst et al. (2000), Monard et al. (2008), Kersanté et al. (2006), etc.
